# Impact of Spino-Pelvic Parameters on the Prediction of Lumbar and Thoraco-Lumbar Segment Angles in the Supine Position

**DOI:** 10.3390/jpm12122081

**Published:** 2022-12-17

**Authors:** Philipp Schenk, Arija Jacobi, Carolin Graebsch, Thomas Mendel, Gunther Olaf Hofmann, Bernhard Wilhelm Ullrich

**Affiliations:** 1Department of Science, Research and Education, BG Klinikum Bergmannstrost Halle gGmbH, 06112 Halle, Germany; 2Department of Orthopedic and Trauma Surgery, DIAKO Ev. Diakonie-Krankenhaus gGmbH, 28239 Bremen, Germany; 3Department of Trauma and Reconstructive Surgery, BG Klinikum Bergmannstrost Halle gGmbH, 06112 Halle, Germany; 4Department of Trauma, Hand and Reconstructive Surgery, Jena University Hospital, Friedrich Schiller University Jena, 07747 Jena, Germany

**Keywords:** spinal alignment, sagittal profile, mono-segmental EPA, bi-segmental EPA

## Abstract

Background: The correction of malposition according to vertebral fractures is difficult because the alignment at the time before the fracture is unclear. Therefore, we investigate whether the spinal alignment can be determined by the spino-pelvic parameters. Methods: Pelvic incidence (PI), pelvic tilt (PT), sacral slope (SS), adjacent endplate angles (EPA), age, sex, body weight, body size, BMI, and age were used to predict mono- and bisegmental EPA (mEPA, bEPA) in the supine position using linear regression models. This study was approved by the Ethics Committee of the Medical Association of Saxony-Anhalt Germany on 20 August 2020, under number 46/20. Results: Using data from 287 patients, the prediction showed R^2^ from 0.092 up to 0.972. The adjacent cranial and caudal EPA showed by far the most frequently significance in the prediction of all parameters used. Anthropometric and spino-pelvic parameters showed sparse impact, which was frequently in the lower lumbar regions. On average, a very good prediction was found. For two mEPA (L3/4 R^2^ = 0.914, L4/5 R^2^ = 0.953) and two bEPA (L3 R^2^ = 0.899, L4 R^2^ = 0.972), the R^2^ was >0.8. However, the predicted EPA differed for individual patients, even in these very effective prediction models—roughly around ±10° as compared to the measured EPA. Conclusions: In general, the prediction showed good to perfect results. In the supine position, the spinopelvic and anthropometric parameters show sparse impact on the prediction of mEPA or bEPA.

## 1. Introduction

Approximately 80% of spinal fractures are located in the lumbar or thoracic sections of the spine, of which 50% are located at the thoracolumbar junction (Th11-L2) [1,2]. Spine fractures can lead to a malalignment, and thus to kyphotic deformity of the spine. When surgical therapy is indicated, one of the surgeon’s goals is to correct the fracture-related malalignment. Achieving the former spinal alignment may be important to avoid static misload and, inter alia, the consequent development of pain. The goal of restoring the physiological spinal alignment can be difficult to achieve due to missing pre-trauma radiological images of the patient’s spine. 

There are efforts and investigations to elucidate the sagittal profile of the spine using pelvic parameters [3,4,5]. The most important parameters are: pelvic incidence (PI), pelvic tilt (PT), and sacral slope (SS). These were primarily determined using X-rays in a standing position [6,7,8]. Correlations between spino-pelvic parameters and lumbar lordosis, as well as between age and the spino-pelvic parameters, could be found [4,9,10].

However, it is unclear whether there is a relationship between the sagittal spine profile and the pelvic parameters, even in the supine position, and whether the pelvic parameters are suitable for estimating the sagittal spine segment angles.

Therefore, the aim of this study is to investigate whether the spinal alignment, captured by the mono- and bi-segmental sagittal Cobb angles, can be determined by the spino-pelvic parameters and the Cobb angles of the adjacent segments.

## 2. Materials and Methods

This study is a retrospective analysis of cross-sectional data, and it was conducted in accordance with the Declaration of Helsinki and approved by the Ethics Committee of the Medical Association of Saxony-Anhalt Germany on 20 August 2020, under number 46/20. Initially, polytrauma CT scans of 1826 patients were examined and evaluated for suitability according to the inclusion and exclusion criteria. The CT images were acquired using the Aquilion PRIME (Canon medical systems Europe) in the supine position. Included in this study were patients with intact spine, pelvis and femur. Patients under 18 years old, with spine deformity or spondylolisthesis with a Meyerding grade greater than 1 [11]; spine, pelvic, or femur fracture, or previous spinal surgery, were excluded, leading to a sample size of 287 patients. 

### 2.1. Parameters

The following spino-pelvic parameters were determined in the CT images: pelvic incidence (PI), pelvic tilt (PT), and sacral slope (SS). The spino-pelvic parameters were measured according to the specifications of Roussouly et al. and Le Huec et al. [3,4].

The PI was measured as the angle between the line perpendicular to the midpoint of the cranial endplate of the sacrum (white dotted line in Figure 1) and the line connecting the midpoint of the sacral plate to the center of one femoral head (red/white dotted line). The PT was measured between the line connecting the midpoint of the cranial endplate of the sacrum to the center of one femoral head and the coronal plane (red solid line). Thus, the PI describes the inclination of the sacrum in relation to the femur head, and the PT indicates the inclination of the sacrum in relation to the body plumb line, which is usually used in the standing position. The SS was measured as the angle between the midpoint of the cranial endplate of the sacrum and the body’s transversal plane (black bold lines in Figure 1). The transversal plane stands perpendicular to the body plumb line. For a better understanding, Figure 1 shows how the measurement of the spino-pelvic parameters was carried out.

In addition to the spino-pelvic parameters, the sagittal vertebral Cobb angles between the cranial and caudal endplates were mono-segmentally (mEPA) measured for each segment from Th8/Th9 to L4/L5 and bi-segmentally (bEPA) for the segments of Th8 to L4, respectively. The mEPA is the angle between the cranial endplate of the cranial vertebra and the caudal endplate of the caudal vertebra of the segment. The bEPA is the angle between the cranial endplate of the nearest cranial vertebra and the caudal endplate of the adjacent caudal vertebra (e.g., bEPA Th12 represents the angle between the cranial endplate Th11 and the caudal endplate L1). However, the procedure for the prediction of the caudal segments of the lumbar spine L3/5, L4/5 (mEPA) and L3, L4 (bEPA) were different due to the issue that no complete mEPA or bEPA could be measured as described above. Therefore, the segment angles of L4/S1 and L5/S1 were used as caudal segments for the prediction and were measured between the cranial endplates of L4 or L5 and S1, respectively.

Lordotic angles are described by positive values and kyphotic angles by negative values.

### 2.2. Statistical Analyses

To identify differences between older and younger patients, the cohort was divided in older and younger patients, according to the median age. Differences regarding each mEPA and bEPA between the age groups were examined using a linear general model for repeated measures. The mEPA and bEPA of the segments were used for repeated measurements. The impact of sex, body weight, body size, BMI, and spino-pelvic parameters (PT, PI, and SS) on the segmental angles were adjusted by using them as covariates. Differences in spino-pelvic parameters were examined using a multivariate general linear model. Sex, body weight, body size, and BMI were used as covariates. Sex was coded with 1 for men and 2 for women.

The results of the Bonferroni-Holm post hoc pairwise comparison between the younger and older group regarding mEPA and bEPA for each segment, as well as the spino-pelvic parameters, are reported as means, standard deviation (sd), lower and upper limit (LL, UL) of 0.95 confidence interval (CI), and minima and maxima (min, max).

To predict the mono- and bisegmental angles from Th9/10 to L4/5 (mEPA) and Th10 and L4 (bEPA), linear regression models were used for each segmental level, respectively. 

For the prediction of mEPA, the adjacent cranial and caudal segments were chosen in the following way. For example, if the mEPA of Th10/11 is the focus for the prediction, it can be predicted using Th8/9 and Th11/12, or using Th9/10 and Th12/L1 as the cranial and caudal segments, respectively. In other words, Th8/9 and Th11/12 can be used as the adjacent mEPA for the prediction of the mEPA of Th9/10 and Th10/11, respectively. Figure 2 shows an exemplary overview of the EPA used for the prediction. In addition to the adjacent EPS, the following variables were used in the regression model: the spino-pelvic parameters (PT, PI, and SS), age, sex, body weight, body size, and BMI.

In an iterative process, the dependent variables were successively reduced to such an extent that all variables ultimately used in the regression model showed significance. 

Corrected R^2^ was used to assess model quality of the linear regression. Model quality with an R^2^ greater than or equal to 0.8 was set to define the prediction as valid. Predictions of valid EPA were subsequently compared with the measured EPA, using difference plots. Therefore, the measured EPAs were used, and the difference between the predicted and the measured EPAs was calculated. The limits of agreement were set using a 1.96-fold standard deviation of the differences between both methods. Linear regression lines were fitted to this data in order to reveal the tendency impact, depending on the values. SPSS V.28 (IBM Corp. Released 2021. IBM SPSS Statistics for Windows, Armonk, NY, USA: IBM Corp) was used for statistical analyses, with *p* = 0.05.

## 3. Results

In total, the data of 208 male and 79 female adults were be included in this study. Due to the median split (40 years), a group of 143 younger adults (28 ± 6 years, range: 18–39 years) and a group of 144 older adults (56 ± 10 years, range: 40–87 years) was generated. 

The younger group included significantly more males (79%, male/female: 113/90) compared to the older group (66%, male/female: 95/49, *p* = 0.017).

In general, no differences could be found for mEPA between the younger and older group (*p* = 0.302). The mEPA differed significant between the segmental levels (*p* < 0.001), except between L4/L5 and L5/S1 (*p* = 0.121), Th12/L1 and Th8/9 (*p* = 1.000), Th9/10 (*p* = 0.660), Th10/11 (*p* = 1.000), between Th11/12 and Th8/9 (*p* = 1.000), and between Th10/11 and Th8/9 (*p* = 1.000). The covariates showed a significant effect for sex (*p* < 0.001), PT (*p* = 0.015), and SS (*p* = 0.009). Body weight (*p* = 0.584), body size (*p* = 0.389), BMI (*p* = 0.509), and PI (*p* = 0.051) showed no significant effect.

For the bEPA, similar results can be found for the main effects regarding age group (*p* = 0.469) and the segments (*p* < 0.001). No significant differences in bEPA could be found between Th10, Th11, and Th12 (*p* > 0.535). Sex was the only covariate which showed significance (*p* = 0.001). The other covariates showed no significance (body weight: *p* = 0.483, body size: *p* = 0.414, BMI: *p* = 0.470, PI: *p* = 0.131, PT: *p* = 0.236, SS: *p* = 0.113).

The multivariate general linear model showed no differences between age groups for PT (*p* = 0.188), PI (*p* = 0.725), and SS (*p* = 0.121). The covariates showed no significant effects on spino-pelvic parameters (sex: *p* > 0.205, body weight: *p* > 0.233, body size: *p* > 0.121, and BMI: *p* > 0.282). Descriptive data for mEPA, bEPA, the spino-pelvic parameters, and the results of the pairwise comparison of the younger and older group are given in Table 1.

Using the iterative process to reduce the number of variables to predict mEPAs using the linear regression with only significant variables led to the exclusion of body weight, body size, BMI, and SS. None of these showed a significant impact on explaining the variance. The coefficients and the model quality of the logistic regression are shown in Table 2. The adjacent mEPA used showed the most frequent impact on the prediction. The model quality for mEPA prediction ranged from R^2^ 0.092 (L2/3) up to 0.953 (L4/5). Valid model quality (R^2^ ≥ 0.8) was reached for the prediction of L3/4 (using L1/2 and L4/5 R^2^ = 0.867, using L2/3 and L5/S1 R^2^ = 0.914) and L4/5 (R^2^ = 0.953). The closest prediction, which failed the validity threshold of 0.8, was L2/3, with an R^2^ of 0.712.

It can be observed that when predicting the mEPA, for nearly every case, the cranial and caudal mEPA had a significant impact. Age showed significance in just two cases, and seemed to have a low impact. Only in the caudal segments of the spine (L3/4 and L4/5), sex, PT, and PI showed significant impacts. 

For bEPA the body weight and BMI showed no significant impact on the prediction. The adjacent bEPA showed the most frequently significant impact. The model quality ranged from 0.294 (L1) up to 0.972 (L4). The prediction of L3 (R^2^ = 0.899) and L4 (R^2^ = 0.962) met the validity criteria of R^2^ ≥ 0.8. For both predictions, the adjacent caudal bEPA showed significance, while the cranial bEPA did not.

For L3 and L4, an excellent model quality could be found with results of 0.899 and 0.972. In contrast to the mEPA, the predicted bEPA showed significant consideration of body height and SS. For both, predicting mEPA and bEPA for L3 and L4, sex showed a significant impact. However, the spino-pelvic parameters showed significance for the prediction of mEPA in just 5 of 42 possible cases (12%) and for the prediction of bEPA, significance was shown in only 6 of 21 possible cases (29%), respectively.

A visual comparison of the predicted (with model quality R^2^ ≥ 0.8) and the measured mEPA (L3/4 and L4/5) and bEPA (L3 and L4), shown as difference plots, is given in Figure 3. For all five predictions, the mean difference is close to zero, which is considered perfect. Nonetheless, the limits of agreement span over a considerable area of roughly ±10°. The linear regression showed a significant trend for each prediction, but with varying degrees. The linear regression of bEPA L4 was very low, with an R^2^ of 0.07. However, the direction of the general trend of the linear regression is comparable in each figure, showing a positive trend. Visually, the scatter clouds show that the variability of the prediction seems not to be dependent on the value of the predicted EPA.

## 4. Discussion

The aim of this study was to investigate if spinal segment angles of the thoraco-lumbar and lumbar spine can be sufficiently predicted using spino-pelvic parameters. In general, the predictions showed good to perfect results. However, this only applies to the generalization of our results. Limiting the threshold of the model, as we have done with a model quality greater than 0.8, resulted in only two or three predictions being supported, depending on whether mono- or bisegmental angles were predicted, respectively. Surprisingly, the spino-pelvic parameters had little influence on the predictions, with the only effect found in the lumbar region. The measurement of the cranial endplate of S1 is mandatory to determine spino-pelvic parameters. In other words, segments that are directly related to S1 are the most likely to be influenced by the orientation of the S1 cranial endplate, which could have a direct effect in this study. Here, the orientation of the sacrum, respectively S1, seems to have direct impact on the segment angles in the lower lumbar segments. Segments that are more cranial seem to be influenced indirectly by the pelvic alignment.

In several studies, PI was found to be the key factor in sagittal balance [8,10,12,13]. In our study, regarding the prediction of single segmental angles, the PI shows significance only in the caudal region of the lumbar spine. The studies mainly examine the influence of the spino-pelvic parameters on the general alignment, i.e., on the thoracic kyphosis and/or lumbar lordosis. The prediction of lumbar lordosis based on spino-pelvic parameters has previously been shown in two studies [8,14]. In addition, in these studies, the parameters are determined in the standing position. There is no consideration of their influence on individual the segment angles.

As for the PI, only a rare impact of the PT in the segmental angles was found. Significance was found for the last two segments of the lumbar spine (mEPA and bEPA), with negative coefficients found for PT and positive coefficients for PI, respectively. The coefficients, and thus, the actual impact on the prediction, is distinctly stronger for the PT than for the PI.

For the PI, the influence on the spinal profile, and thus, on these segments, is easy to imagine. The more the sacrum is tilted, the greater the PI, the greater the lordosis, and thus, the larger the individual segment angles. The less tilted it is, the more it is parallel to the horizontal plane in supine position. 

Regarding the geometric similarity for the PT, the more the pelvis is retro verted, the higher the PI and the more pronounced the lordosis.

Our data and analyses indicate that the closer the segments are located to the sacrum, the stronger the impact of the PT and the PI on the segment angles. In the cranial segments of the spine, their influence seems to completely disappear. For the SS, we would have assumed a similar relationship, compared to the PT and PI. However, no significant impact on the prediction can be determined for the monosegmental segments. 

We assume that the influence of the three spino-pelvic parameters and their effects interact and thus, cannot be reduced at a one-dimensional level. Additionally, other variables, including the position in which the parameters are determined, may interact and affect the segmental angles [15]. 

In our prediction, we considered body weight, height, and BMI as anthropometric. However, none of these variables were found to affect the segment angles in the supine position. 

The body weight, and thus, the BMI, have an effect on the spino-pelvic alignment in the standing position [16,17,18]; however, their impact on segment angles was not examined here. Therefore, it is unclear whether their impact on the individual segment angles exists in the standing position, or whether the supine position is the reason for the disappearance of this impact. Of course, it may be that the effect on the individual segment angles is quite small, and that this effect only becomes apparent when considering the thoracic kyphosis or lumbar lordosis. 

Age and gender show only a rare significance. These findings affect the predictions of the mEPA and bEPA in a comparable manner. 

In investigating the effect of age on the spino-pelvic parameters, no differences could be found between age groups. Various authors have so far, been able to demonstrate age effects on the spino-pelvic parameters [9,12,17,19]. 

Different authors have been able to show differences between men and women with regard to the spino-pelvic parameters in the standing position [13,17,20]. Our study is based on data collected from patients in the supine position. We suggest that this is the main reason that we did not find comparable results. However, our surgeons assessed the extent of fracture reduction in the patient in CT images collected in the supine position. 

Naturally, effects of positional changes—standing versus supine—are to be expected. Radiological diagnosis is also possible in the standing position, but this is not possible for every patient, depending on the injury. Additionally, not every hospital has the technical equipment for this type of diagnostic method. 

It would be interesting to investigate the prediction of mEPA and bEPA for a population in which radiological diagnostics were performed in both a standing and a supine position in order to compare the results of both predictions. Nevertheless, the consideration of spino-pelvic parameters in fracture correction is recommended [21]. Furthermore, the spino-pelvic parameters also show an impact on the patient’s quality of life [9], and therefore, should be of particular consideration.

In this respect, the correction or non-correction of fractures not only has an impact on the spino-pelvic alignment, but also has a secondary effect on the patient’s quality of life.

In this study, we did not examine lumbar lordosis per se. The aim of this study was to determine valuable parameters to predict segment angles. These parameters should not have a great deal of influence if a fracture has occurred. In a fractured spine, it can be assumed that depending on the severity and location of the fracture, the thoracic kyphosis or lumbar lordosis might have changed, and it is likely unclear how the fracture has affected the other segmental angles. Differences between mEPA or bEPA in the lower lumbar region are greater than in the more cranial segment angles, respectively. This might certainly be a reason why there is a better prediction in the lumbar area, and why the influences of the adjacent segment angles are greater here. In the more cranial regions, the differences in segment angles between may be too small to differentiate between segments.

If the surgeon orientates the fracture reduction on the lumbar lordosis from S1 to the inflection point, however, in our experience, the inflection point appears to be more cranial when lying down than when in a standing position, which makes lumbar lordosis less suitable as a target parameter in the supine position.

During the surgical reduction of the fracture, the avoidance of insufficient or excessive reduction of fractures, lacking sufficient information regarding the individual physiological profile, is also in surgical focus. With precise individual adjustment of the fracture, less additional compensation may be necessary, resulting in better rehabilitation after surgery. Particularly after spondylodesis and existing malalignment, unfavorable loads on the adjacent movement segment are to be expected. Misalignments due to poor fracture position likely diminish the success of subsequent rehabilitation and can lead to pain, adjacent fracture, or other complications. The combination of nutritional supplements and rehabilitation programs can reduce the patient’s pain and improve the quality of life [22]. However, spinal alignment is not the only thing that determines recovery. Even with poor posture, the health of patients can be positively influenced by training [23].

Finally, only a few of the parameters used in this study are suitable for predicting segment angles in general. Overall, the caudally and cranially adjacent segment angles seems to be sufficient parameter for use in predicting the segment angles between them. The prerequisite for this, however, is that the adjacent segments are intact. Looking closer at our results, the predictions achieve a wide range, which is shown by the relatively large limits of agreement in the difference plots. Regarding the mean values of the mEPA and bEPA, plus/minus 10 degrees is too great to provide a good individual recommendation for the scope of correction. Ultimately, further investigations must be carried out in order to predict the segment angles in the supine position, considering the spino-pelvic parameters. Questions also arise regarding the amount of malalignment for which the human body can physiology compensate, as well as the way in which this compensation changes with age. 

## Figures and Tables

**Figure 1 jpm-12-02081-f001:**
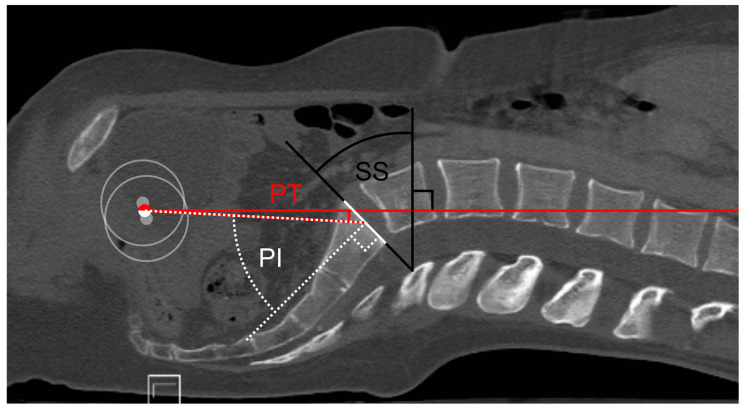
Measurement of the spino-pelvic parameters. The head of both femurs (grey circles) and their mean is marked with a white/red circle. A horizontal red solid line is drawn crossing the mean of the femur heads. The cranial endplate of the first sacral vertebra is highlighted with a white solid line. The pelvic tilt (PT) was measured as the angle between the red horizontal line and the white/red dotted line between the mean of the femur heads and the center of the cranial endplate of S1. The angle of the pelvic incidence (PI) was measured as the angle between the white/red dotted line between the mean of the femur heads and the center of the cranial endplate of S1, perpendicular to the midpoint of the cranial endplate of S1 (white dotted line). The sacral slope (SS) was measured as the angle between the cranial endplate of S1 perpendicular to the horizontal red solid line, highlighted with black solid lines.

**Figure 2 jpm-12-02081-f002:**
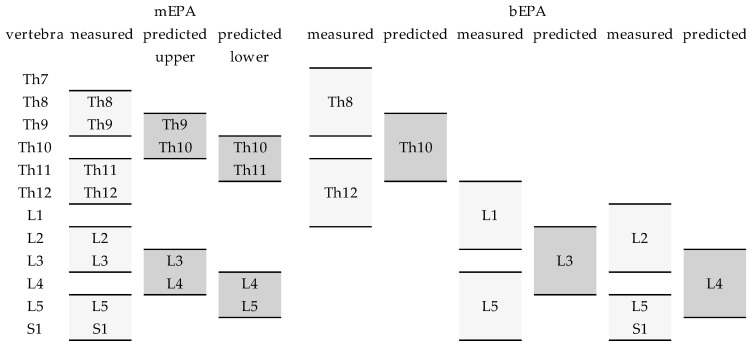
Schematic and exemplary representation of the measured (light grey) and the predicted (dark gray) monosegmental (mEPA) and bisegmental endplate angles (bEPA). The EPAs were measured as the enclosing angles of the cranial endplate of the cranial vertebra and the caudal endplate of the caudal vertebra. With one exception at S1 (first vertebra of the sacrum), here the cranial endplate was used as the caudal end of the EPA. The further cranial and caudal EPAs are shown as examples. The setup was chosen to predict the EPA in fractures of single vertebrae from Th10 to L4. The second example for bEPA, predicting L3, uses the bEPA of L1 and the EPA of the cranial endplates of L4 and S1. Predicting the bEPA of L4, the bEPA of L2 and the EPA of the cranial endplates of L5 and S1 were used.

**Figure 3 jpm-12-02081-f003:**
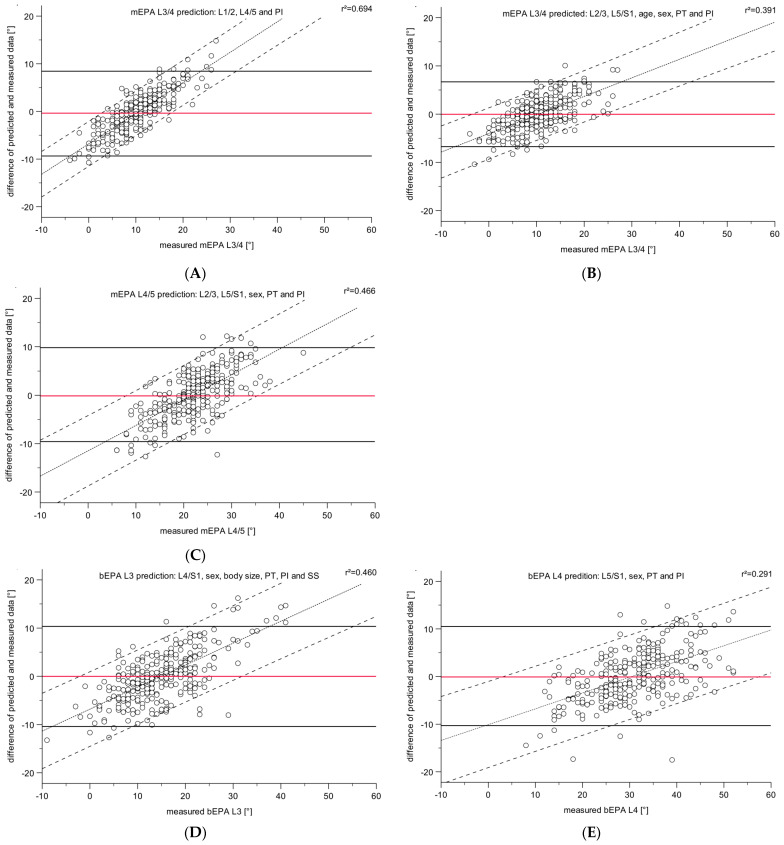
Difference plots of the measured mEPA ((**A**): L3/4, (**B**): L3/4, (**C**): L4/5) and bEPA ((**D**): L3, (**E**): L4) and their difference from the predicted monosegmental (mEPA) and bisegmental endplate angles (bEPA), respectively. Horizontal lines are the limits of agreement (black) and the mean difference (red); dotted and dashed lines are the regression line and their 0.95% confidence intervals. To detect the dependency of differences between measured and predicted angles, R^2^ is given in the upper right corner of each plot.

**Table 1 jpm-12-02081-t001:** Angles of thoraco-lumbar mono- and bisegmental EPA and spino-pelvic parameters (PT: pelvic tilt, PI: pelvic incidence, SS: sacral slope) of younger and older adults. Data are given as mean and standard deviation, the lower (LL) and upper limits (UL) of the 95% confidence interval (CI), minimum and maximum, and p-value of pairwise comparison.

		Younger than 40 Years (N = 143)				40 Years and Older (N = 144)				
		Mean ± Sd	0.95 CI (LL UL)	Min	Max	Mean ± Sd	0.95 CI (LL UL)	Min	Max	*p* Value
monosegmental	Th8/9	−4.3 ± 4.3	(−5.0 −3.5)	−24	3	−5.0 ± 4.2	(−5.6 −4.3)	−15	4	0.164
	Th9/10	−2.8 ± 4.2	(−3.5 −2.1)	−18	6	−3.0 ± 3.9	(−3.7 −2.4)	−13	8	0.607
	Th10/11	−4.5 ± 3.8	(−5.1 −3.9)	−16	5	−4.2 ± 4.0	(−4.9 −3.6)	−17	7	0.510
	Th11/12	−5.3 ± 4.0	(−5.9 −4.6)	−17	5	−5.4 ± 4.7	(−6.1 −4.6)	−16	6	0.842
	Th12/L1	−4.4 ± 4.5	(−5.1 −3.6)	−16	12	−3.4 ± 4.6	(−4.1 −2.6)	−13	11	0.058
	L1/L2	−1.3 ± 4.9	(−2.1 −0.5)	−16	13	−0.3 ± 5.2	(−1.2 0.5)	−11	13	0.102
	L2/3	4.1 ± 5.1	(3.2 4.9)	−6	21	5.0 ± 4.9	(4.2 5.8)	−8	19	0.120
	L3/4	10.3 ± 5.5	(9.4 11.2)	−4	26	10.4 ± 5.6	(9.5 11.4)	−2	27	0.877
	L4/L5	22.4 ± 6.7	(21.3 23.5)	6	45	21.8 ± 6.2	(20.8 22.8)	6	38	0.389
	L5/S1	23.9 ± 6.0	(22.9 24.9)	6	50	23.7 ± 6.9	(22.6 24.9)	0	49	0.814
bisegmental	Th10	−6.0 ± 4.8	(−6.8 −5.2)	−21	5	−6.1 ± 4.9	(−6.9 −5.3)	−17	7	0.866
	Th11	−6.7 ± 4.7	(−7.5 −6.0)	−23	6	−6.6 ± 5.4	(−7.5 −5.7)	−22	5	0.856
	Th12	−6.7 ± 5.4	(−7.6 −5.8)	−21	11	−6.2 ± 5.7	(−7.2 −5.3)	−19	11	0.489
	L1	−2.3 ± 6.0	(−3.3 −1.4)	−16	16	−1.2 ± 6.3	(−2.2 −0.1)	−14	18	0.101
	L2	4.1 ± 7.0	(2.9 5.2)	−10	27	6.0 ± 7.0	(4.8 7.1)	−10	27	0.024
	L3	14.2 ± 7.8	(12.9 15.5)	−9	41	15.6 ± 7.9	(14.3 16.9)	0	41	0.144
	L4	30.7 ± 9.1	(29.2 32.2)	11	52	30.9 ± 8.2	(29.6 32.3)	8	51	0.837
	PT	8.8 ± 6.8	(7.7 9.9)	−7	29	10.0 ± 6.0	(9.0 11.0)	−4	26	0.116
	PI	50.5 ± 11.0	(48.7 52.3)	27	86	52.7 ± 10.6	(50.9 54.4)	28	86	0.092
	SS	41.9 ± 8.0	(40.6 43.3)	21	71	42.7 ± 7.7	(41.5 44)	18	73	0.390

**Table 2 jpm-12-02081-t002:** Results of the linear regression model for the prediction of mono- and bisegmental endplate angles (mEPA, bEPA). The explained variance (R^2^) and the significant coefficients (ExpB) for the parameters used are given. Sex was coded as 1 for men and 2 for women.

		EPA Used			ExpB							
	Predicted	Cranial	Caudal	R^2^	Cranial	Caudal	Age	Sex	Body Size	PT	PI	SS
monosegmental	Th9/10	Th8/9	Th11/12	0.568	0.597 ^A^							
	Th10/11	Th8/9	Th11/12	0.645	0.334 ^A^	0.472 ^A^						
	Th10/11	Th9/10	Th12/L1	0.578	0.698 ^A^	0.345 ^A^						
	Th11/12	Th9/10	Th12/L1	0.626	0.419 ^A^	0.760 ^A^						
	Th11/12	Th10/11	L1/2	0.581	0.867 ^A^	0.232 ^A^						
	Th12/L1	Th10/11	L1/2	0.523	0.485 ^A^	0.613 ^A^						
	Th12/L1	Th11/12	L2/3	0.576	0.634 ^A^	0.229 ^A^	−0.028 ^B^					
	L1/2	Th11/12	L2/3	0.295	0.404 ^A^	0.457 ^A^						
	L1/2	Th12/L1	L3/4	0.425	0.661 ^A^	0.181 ^A^						
	L2/3	Th12/L1	L3/4	0.712	0.245 ^A^	0.539 ^A^						
	L2/3	L1/2	L4/5	0.092	0.405 ^A^							
	L3/4	L1/2	L4/5	0.867	0.215 ^A^	0.367 ^A^					0.050 ^B^	
	L3/4	L2/3	L5/S1	0.914	0.426 ^A^	−0.295 ^A^	−0.033 ^B^	2.718 ^A^		−0.393 ^A^	0.332 ^A^	
	L4/5	L2/3	L5/S1	0.953		−0.316 ^A^		4.745 ^A^		−0.729 ^A^	0.588 ^A^	
bisegmental	Th10	Th8	Th12	0.696	0.492 ^A^	0.233 ^A^						
	Th11	Th9	L1	0.548	0.698 ^A^	0.436 ^A^						
	Th12	Th10	L2	0.413	0.867 ^A^	0.258 ^A^						
	L1	Th11	L3	0.294	0.320 ^A^	0.370 ^A^			−0.029 ^A^			
	L2	Th12	L4	0.611	0.335 ^A^	0.236 ^A^	0.054 ^B^		−0.075 ^A^			0.255 ^A^
	L3	L1	L4/S1	0.899		−0.576 ^A^		5.005 ^A^	−0.047 ^A^	−0.648 ^A^	0.533 ^B^	0.414 ^B^
	L4	L2	L5/S1	0.972		−0.611 ^A^		6.135 ^A^		−1.140 ^A^	0.933 ^A^	

^A^: *p* < 0.001. ^B^: *p* < 0.05.

## Data Availability

The data from all subjects involved and analyzed in this study are available from the corresponding author on reasonable request.

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
