# Peer review of "Impact of Spino-Pelvic Parameters on the Prediction of Lumbar and Thoraco-Lumbar Segment Angles in the Supine Position"

_jpm, 2022, doi:10.3390/jpm12122081_

Round 1

Reviewer 1 Report

Dear authors,

I was pleased to review the paper entitled " Impact of spino-pelvic parameters on the prediction of lumbar and thoraco-lumbar segment angles in supine position. “

What is the main question addressed by the research? Is it relevant and interesting? How original is the topic?

The analysis of spinopelvic parameters is currently one of the major fields of interest in spine surgery. It is very difficult to evaluate the restoration of these parameters in the supine position. Particularly this study investigate if the spinal alignment can be determined by the spino-pelvic parameters.

Are the conclusions consistent with the evidence and arguments presented? Do they address the main question posed? Is the text clear and easy to read? Is the paper well written?

- introduction:

Well written, concise.

- Material and Methods:

Well written with good Statistical analyses

Result

Well written, concise. Conclusions are consistent with

the evidence and arguments presented

Discussion

Well written, concise

Author Response

Dear Reviewer,

thank you for revising our manuscript „ Impact of spino-pelvic parameters on the prediction of lumbar and thoraco-lumbar segment angles in supine position”.

We are very pleased that the manuscript resonated with you and that we were able to generate understanding for this topic. Thank you very much and we wish you a nice and healthy time.

On behalf of all authors.

Reviewer 2 Report

Dear Sirs,

Your paper presents some aspects that have to be addressed.

ABSTRACT: please remove headings, according to the Journal editorial rules.

INTRODUCTION: concise, it well described background and aims of the study.

MATERIALS: please correct the layout of the text at lines 80-84.

Which is the study design? This aspect is not clear. Please describe it both in the abstract and in this section and report ethical approval number, as You did in the statements at the end of the paper.

DISCUSSION: this section must be improved. It should go beyond the narrow surgical technical aspects of spine assessment parameters, explaining also the possible consequences of your findings from a clinical and rehabilitative point of view. Could these findings help to prevent spine fractures and to better target rehabilitation projects? To do that, I suggest the following references:

Farì, G., Santagati, D., Pignatelli, G., Scacco, V., Renna, D., Cascarano, G., Vendola, F., Bianchi, F. P., Fiore, P., Ranieri, M., & Megna, M. (2022). Collagen Peptides, in Association with Vitamin C, Sodium Hyaluronate, Manganese and Copper, as Part of the Rehabilitation Project in the Treatment of Chronic Low Back Pain. Endocrine, metabolic & immune disorders drug targets22(1), 108–115. https://doi.org/10.2174/1871530321666210210153619

 - Notarnicola, A., Farì, G., Maccagnano, G., Riondino, A., Covelli, I., Bianchi, F. P., . . . Moretti, B. (2019). Teenagers’ perceptions of their scoliotic curves. an observational study of comparison between sports people and non- sports people. Muscles, Ligaments and Tendons Journal, 9(2), 225-235.

Best regards
